# Acclimation of Hairless Spontaneously Hypertensive Rat to Ambient Temperature Attenuates Hypertension-Induced Pro-Arrhythmic Downregulation of Cx43 in the Left Heart Ventricle of Males

**DOI:** 10.3390/biom14121509

**Published:** 2024-11-26

**Authors:** Katarina Andelova, Matus Sykora, Veronika Farkasova, Tatiana Stankovicova, Barbara Szeiffova Bacova, Vladimir Knezl, Tamara Egan Benova, Michal Pravenec, Narcis Tribulova

**Affiliations:** 1Centre of Experimental Medicine, Slovak Academy of Sciences, 841 04 Bratislava, Slovakia; katarina.andelova@savba.sk (K.A.); matus.sykora@savba.sk (M.S.); veronika.farkasova@savba.sk (V.F.); barbara.bacova@savba.sk (B.S.B.); exfaknem@savba.sk (V.K.); usrdtabe@savba.sk (T.E.B.); 2Department of Pharmacology and Toxicology, Faculty of Pharmacy, Comenius University in Bratislava, 832 32 Bratislava, Slovakia; stankovicova@fpharm.uniba.sk; 3Institute of Physiology, v.v.i., Academy of Sciences of the Czech Republic, 14220 Prague, Czech Republic; michal.pravenec@fgu.cas.cz

**Keywords:** hairless SHR, males, females, left and right heart ventricle, connexin43, arrhythmias

## Abstract

Objectives: Due to poor treatment adherence and lifestyle-based interventions, chronic hypertension is a dominant risk factor predisposing individuals to heart failure and malignant arrhythmias. We investigated the impact of the postnatal acclimation of hairless SHR to ambient temperature that is, for them, below thermoneutrality, on the electrical coupling protein connexin-43 (Cx43) and pro-fibrotic markers in both heart ventricles of male and female hairless SHR rats compared to the wild SHR. Methods: Some 6-month-acclimated male and female hairless SHR as well as age- and sex-matched wild SHR were included and compared with the non-hypertensive Wistar strain. The left and right heart ventricles were examined for Cx43 topology, myocardial structure, and the histochemistry of capillaries. The protein levels of Cx43, relevant protein kinases, and extracellular matrix proteins (ECMs) were determined by immunoblotting. MMP-2 activity was assessed via zymography, and susceptibility to malignant arrhythmias was tested ex vivo. Results: Cx43 and its phosphorylated variant pCx43^368^ were significantly reduced in the left heart ventricles of wild SHR males, while to a lesser extent in the hairless SHR. In contrast, these proteins were not significantly altered in the right heart ventricles of males or in both heart ventricles in females, regardless of the rat strain. Pro-arrhythmic Cx43 topology was detected in the left heart ventricle of wild SHR and to a lesser extent in hairless SHR males. TGFβ protein was significantly increased only in the left ventricle of the wild SHR males. MMP-2 activity was increased in the right ventricle but not in the left ventricles of both males and females, regardless of the rat strain. Conclusions: The findings indicate that the postnatal acclimation of hairless SHR to ambient temperature hampers the downregulation of Cx43 in the left heart ventricle compared to wild SHR males. The decline of Cx43 was much less pronounced in females and not observed in the right heart ventricles, regardless of the rat strain. It may impact the susceptibility of the heart to malignant arrhythmias.

## 1. Introduction

The prevalence of essential hypertension (HTN) in the general population contributes to a major global health issue [1], associated with an increased burden in mortality due to heart failure and the occurrence of life-threatening arrhythmias [2,3,4]. High systolic blood pressure and left ventricular hypertrophy are established independent predictors of cardiovascular diseases (CVDs) [5]. Despite the availability of antihypertensive treatments, almost half of the patients fail to achieve the recommended values outlined in the guidelines [6]. Poor adherence to lifestyle modifications and resistance to antihypertensive drugs are contributing factors [7]. The high incidence of CVD and the increased propensity of the heart to malignant arrhythmias associated with chronic HTN [5,8] challenge precision medication [7] and highlight the need for further molecular research to reveal novel targets for protection.

Compelling data from spontaneously hypertensive rat strain (SHR) mimicking human essential HTN [8] indicate that the pressure overload-induced remodeling of cardiac myocytes (hypertrophy) and extracellular matrix (ECM), including fibrosis, are crucial factors affecting cardiac connexin-43 (Cx43). This remodeling increases the susceptibility of the heart to malignant arrhythmias. The heart is an electro-mechanical pump, and Cx43 channels at the gap junctions at the intercalated disk (ID) are essential for electrical coupling. This coupling allows the propagation of cardiac action potential (AP), which induces the contraction of cardiomyocytes. Impairment in normal AP conduction is a key factor that increases susceptibility to the development of cardiac arrhythmias [9,10]. Robust cardiac myocyte coupling, ensured by Cx43 channels at the ID, is critically important for proper cardiac conduction and protection against life-threatening ventricular fibrillation (VF) [8]. In contrast, the downregulation of Cx43 and its abnormal topology due to hypertrophy and fibrosis disturb myocardial conduction, electrical stability, and synchronized contraction, promoting the occurrence of malignant arrhythmias [11,12].

In this context, it is interesting to note that the anti-arrhythmic protection observed during hypothermia in winter-hibernating versus non-hibernating mammals is associated with improved cardiac gap junction function due to the overexpression of Cx43 [13,14,15]. Mild hypothermia is considered beneficial in preventing CVD, increasing heart ischemic resilience [16,17], and preserving myocardial conduction during ischemia by maintaining Cx43 expression and its topology at the ID [18]. The postnatal acclimation of hairless SHR phenotype to ambient standard chow temperature resulted in metabolic adaptation and increased thermogenesis [19]. Moreover, our previous study indicates the upregulation of myocardial Cx43 and suppression of pro-fibrotic markers in the left heart ventricle of these rats [20]. However, there is a lack of evidence regarding whether HTN and/or “cold” acclimation also affect the right heart ventricle and whether there are possible sex differences.

Taken together, the intention of this study was to investigate Cx43 and ECM proteins in both the left and right ventricles of male and female SHR, as well as hairless SHR, after a 6-month acclimation to ambient temperature. Additionally, relevant protein biomarkers and microscopic examinations were performed. Heart function was assessed using echocardiography and electrocardiography, and the VF threshold was estimated in an ex vivo perfused heart model.

## 2. Materials and Methods

### 2.1. Animals and Experimental Design

The experiments were performed using 6-month-old 27 male and 27 female rats (*n* = 9 per group, i.e., 6 rats for tissue analysis and 3 rats for testing of VF threshold). We used wild-type spontaneously hypertensive rats (wild SHR), hairless SHR (hairless SHR) due to a genetic mutation in desmoglein-4 and Wistar rats as reference normotensive strain. The wild SHR and hairless SHR were obtained from the accredited breeding facility of the Institute of Physiology, v.v.i., Academy of Sciences of the Czech Republic, Prague. The Wistar rats were sourced from an accredited breeding facility—Department of Toxicology and Laboratory Animal Breeding CEM, SAV, v.v.i., ÚEFT, Dobrá Voda. All the rats were housed at standard 22 °C with 12 h light/dark cycles and had ad libitum access to tap water and standard laboratory chow. The hairless SHR strain is characterized by postnatal acclimation to this ambient temperature, which is, for them, under thermoneutrality, resulting in increased adaptive thermogenesis along with metabolic adaptation [19]. The maintenance and handling of the animals were performed in accordance with the “Guide for the Care and Use of Laboratory Animals”, published by the U.S. National Institutes of Health (NIH Publication, 8th ed., revised 2011) and in accordance with European Union Council Directive 86/609EEC. On 13 September 2021, the Ethical Committee of State Veterinary and Food Administration of the Slovak Republic approved the project registered under number 5278-3/2021-220. The experiment was terminated by the euthanasia of the rats with the intraperitoneal administration of Thiopental (VUAB Pharma a.s., Czech Republic) at a dose of 80 mg/kg of body weight. The heart was excised into ice-cold saline, followed by weight registration, and the left and right heart ventricles were sampled, snap-frozen in liquid nitrogen (−160 °C), and stored in a freezer at −80 °C for the subsequent Western blot analysis of targeted proteins, as well as for histological staining, histochemical examinations of the capillary network, and the immunofluorescence detection of Cx43. Additionally, the body weight, heart weight, left and right ventricle weight, and the weight of the brown adipose tissue (BAT) of the rats were registered at the end of the experiment. Systolic blood pressure was measured using a noninvasive plethysmographic method on the caudal artery with the PowerLab 4/30 system (ADInstrument, Budapest, Hungary).

### 2.2. Cardiac Structure and Function Assessed by Echocardiography

Echocardiography was performed 2 days before the end of the experiment using the GE Vivid S6 Dimension echocardiography platform with a 10 MHz linear array transducer (GE-Vingmed Ultrasound AS, Horten, Norway). The rats were anesthetized with isoflurane (Isoflurin, 1000 mg/g, VETPHARMA ANIMAL HEALTH, S.L., Barcelona, Spain) at an initial concentration of 5% and a maintenance concentration of 2%. The examination was conducted using the Vivid E9 XDClear device with a linear phased-array probe ML-6-15-D (GE Healthcare, Boston, NY, USA). To improve image quality, the fur on the chest area of the rats was removed, and an ultrasound transmission gel (Aquasonic 100 Ultrasound Transmission Gel, Parker Labs, Fairfield, NJ, USA) was applied. The functional and structural parameters of the left ventricles were recorded in two-dimensional mode (2D mode) as well as in motion mode (M-mode), and the measured parameters were subsequently averaged from three recordings. The EchoPac software (version 201.71.0, GE Healthcare, USA) was used for the subsequent analysis of the obtained recordings as previously reported [21].

### 2.3. Electrocardiographic Examination

Electrocardiography (ECG) was performed in conscious rats after careful handling before monitoring using a PowerLab converter (Seiva Praktik, Seiva s.r.o., Prague, Czech Republic). Recordings were analyzed with the LabChart system (Seiva Database Veterinary, Seiva, s.r.o. Czech Republic). During the measurement, each limb of the rat was placed on a plate with copper electrodes, onto which a conductive hydrogel (Aquagel-C, Polychem, Czech Republic) was applied. The ECG recording was analyzed; as described previously [22], we recorded 3 standard leads and 3 amplified leads, and analysis was performed using standard II. lead for the duration of the PQ and QT intervals, as well as the duration of the QRS complex. The corrected QT interval (QTc) was calculated using the following formula: QTc = QT/√(RR/200) (QT—QT interval in ms; RR—the time between R waves in ms; 200—a factor accounting for heart rate).

### 2.4. Estimation of VF Threshold Ex Vivo in Langendorff-Mode Perfused Hearts

As previously reported in [23], and illustrated in Figure 1, the hearts of the rats were perfused via a cannulated aorta with oxygenated Krebs–Henseleit solution at a constant pressure of 80 mm Hg (1 mm Hg = 133.322 Pa) and temperature of 37 °C. After 20 min of stabilization, the threshold to induce sustained ventricular fibrillation (VF) (lasting 2 min) was examined. Briefly, 1 s burst of electrical rectangular pulses (100 pulses/s), 1 ms in duration at a current strength of 20 mA (Electrostimulator ST-3, Medicor, Hungary), was delivered via stimulating electrodes attached to the epicardium of the right ventricle. If sustained VF was not induced, the stimulus intensity was increased in 5 mA increments up to a maximum of 50 mA.

### 2.5. Microscopic Examination and Quantitative Image Analysis of Heart Tissue Samples

Conventional hematoxylin–eosin (HE) and Van Gieson (VG) staining of the right and left ventricular sections were performed to detect structural alterations.

Histochemistry was conducted on the cryosections of the right and left ventricles to detect the in situ activity of alkaline phosphatase (AP, E.C.3.1.3.1), a marker of arterial capillaries, and dipeptidyl peptidase-4 (DPP4, E.C.3.4.15.4), a marker of venous capillaries [24]. This was performed to assess the status and the function of the myocardial arterial and venous capillary networks and to reveal microvascular dysfunction. In the quantitative analysis of images from histochemical reactions, positive staining, which corresponds to the activity of individual enzymes, was defined as the area with a pixel count having a threshold value below 128 on the “0–255 gray scale”.

The immunofluorescence labeling of myocardial Cx43 was performed on the right and left ventricular cryosections as previously reported in [20] to detect alterations in myocardial expression and topology. A primary anti-Cx43 antibody (1:700, MAB3068, CHEMICON International, Inc., Temecula, CA, USA) was used, followed by a secondary antibody conjugated with anti-mouse FITC fluorescein isothiocyanate (1:1000, Jackson Immuno Research Labs, West Grove, PA, USA). The quantitative image analysis of the acquired Cx43 images was performed using a Zeiss Apotome 2 microscope (Carl Zeiss, Jena, Germany) on 15 randomly selected areas per section. The area of the positive signal was defined as the number of pixels with a value higher than 128 on the “0–255 gray scale”. Total immunopositivity was expressed as “integral optical density” (IOD) per area. The lateral localization of Cx43 in cardiomyocytes, indicative of its pathophysiological distribution, was quantified following previously established methods [25]. After manually outlining the Cx43 immunolabeling at terminal intercalated disks, the lateral Cx43 was determined by subtracting the IOD of the terminal Cx43 from the total IOD. This lateral Cx43 was expressed as a percentage, calculated by dividing the IOD of lateral Cx43 by the total IOD.

### 2.6. Determination of Myocardial Protein Levels of Cx43, PKCε, PKCδ, TGFb, SMAD2, and MMP-2 by Western Blotting

Frozen left and right ventricular tissues were used for the Western Blot analysis of the following proteins: Cx43 protein and its phosphorylated (p368) variant; protein kinases PKCε and PKCδ, which phosphorylate Cx43; and extracellular matrix proteins—TGFβ, SMAD2, and MMP-2. As described in detail [20], the nonspecific binding of antibodies was blocked with skimmed milk, and the membranes were incubated overnight with the following primary antibodies: anti-total Cx43—1:5000, C6219, Sigma-Aldrich, MI, USA; anti-pCx43^368^—1:1000, sc-101660, Santa Cruz Biotechnology, Dallas, TX, USA; anti-PKCε—1:1000, sc-214, Santa Cruz Biotechnology, TX, USA; anti-PKCδ—1:1000, sc-213, Santa Cruz Biotechnology, Dallas, TX, USA; anti-TGFβ—1:1000, SAB4502954, Sigma-Aldrich, MI, USA; anti-SMAD2—1:1000, #3102, Cell Signaling Technology, Denver, CO, USA; anti-MMP-2—1:1000, sc-10736, Santa Cruz Biotechnology, Dallas, TX, USA; and anti-GAPDH—1:1000, sc-25778, Santa Cruz Biotechnology, TX, USA. This was followed by incubation for 1 h with horseradish peroxidase-linked secondary anti-rabbit antibodies: 1:2000, 7074S, Cell Signaling Technology, Denver, CO, USA or anti-mouse: 1:2000, 7076C, Cell Signaling Technology, Denver, CO, USA. The proteins were visualized using an Amersham Imager 600 (GE Healthcare Bioscience AB, Danderyd, Sweden), and band quantification was performed densitometrically using the Carestream Molecular Imaging Software (version 5.0, Carestream Health, New Haven, CT, USA) and normalized to GAPDH.

### 2.7. Detection of MMP-2 Activity Using Gelatine Zymography

The homogenates of cardiac right and left ventricles were processed as described previously [26]. Protein samples, each containing 40 μg of protein per lane, were loaded into the wells of 10% gels copolymerized with gelatine at a concentration of 2 mg/mL (utilizing Mini-Protean TetraCell equipment from Bio-Rad). These samples were then separated via SDS-PAGE. The gels underwent two rounds of washing for 30 min each with a washing buffer at room temperature (composed of 50 mmol/L Tris-HCl, 2.5% Triton X-100, pH 7.4). Following this, they were left to incubate overnight in a developing buffer at 37 °C (composed of 50 mmol/L Tris-HCl, 10 mmol/L CaCl2, and 1.25% Triton X-100; pH 7.4). Subsequently, staining was performed for one hour using a solution of 1% Coomassie Brilliant Blue G-250 dissolved in a mixture containing 10% acetic acid and 40% methanol. Finally, destaining was carried out using a solution comprising 10% acetic acid and 40% methanol until a distinct differentiation emerged between the transparent bands and the dark blue background. The enzymatic activities of MMP-2 within the transparent bands were then assessed using the Carestream Molecular Imaging Software (version 5.0, Carestream Health, New Haven, CT, USA).

### 2.8. Statistical Evaluation

The Kolmogorov–Smirnov test was applied to verify the normality of the data distribution. Group-related differences were assessed using a one-way analysis of variance (ANOVA) followed by Bonferroni’s multiple comparison test. The results are presented as means ± standard deviations (SD), and a *p*-value of less than 0.05 is considered statistically significant. STATISTICA 10 was used for all the statistical analyses.

## 3. Results

### 3.1. Basic Biometric Parameters

As summarized in Table 1, compared to the normotensive Wistar rats, systolic blood pressure (BP) was significantly elevated in both the wild and hairless SHR males, with a less pronounced increase in females. The body weight (BW) of the hairless SHR males was lower compared to the normotensive Wistar rats. Additionally, in females, both the wild and hairless SHRs exhibited reduced body weights compared to the female Wistar rats. The left ventricular weight was higher in the hypertensive males and females than in the normotensive sex-matched Wistar rats. There were no significant differences in the right ventricular weight among the normotensive and hypertensive rat strains, regardless the sex. Noteworthy, the weight of BAT and BAT/BW ratio was significantly increased in both the male and female hairless SHR compared to the wild SHR and normotensive Wistar rats.

### 3.2. Structural and Functional Heart Parameters Assessed by Echocardiography

As illustrated in Table 2, there was a significant increase in interventricular septal thickness in diastole (IVSd) in the wild SHR males compared to the normotensive Wistar rats. The left ventricular internal diameter in diastole (LVId) and systole (LVIDs) was significantly increased in the wild SHR males compared to the Wistar rats. A decrease in LVIDs was observed in the hairless SHR compared to wild SHR. There were no significant changes in relative wall thickness (RWT) among the experimental rats, regardless of strain or gender. No significant differences in the examined parameters were observed between the female rat strains.

As shown in Table 3, compared to the normotensive male rats, the heart rate (HR) was significantly lower in both the wild and hairless SHRs. There were no significant changes in cardiac output (CO) among the examined rat strains. However, ejection fraction (EF) was reduced in the wild SHR males, but not in the hairless SHR when compared to the normotensive Wistar rats. End-diastolic volume (EDV) was significantly increased in the wild SHR males, but not in the hairless SHR. End-systolic volume (ESV) was increased in the wild SHR males but decreased in the hairless SHR compared to the Wistar rats. There were no significant alterations in the examined parameters among the female rat strains.

### 3.3. Basic Electrocardiographic Parameters

Data were acquired from conscious experimental rats and are summarized in Table 4. There was a significant increase in the heart rate-corrected QT interval (QTc) in the wild SHR, while it was decreased in the hairless SHR compared to the normotensive male Wistar rats. There were no significant alterations among the female rat strains.

### 3.4. Incidence of Sustained Ventricular Fibrillation Assessed in Ex Vivo Perfused Rat Heart

The propensity of the experimental rats to sustained ventricular fibrillation (SVF) induced by electrical stimulation is summarized in Table 5. SVF was not induced in the male nor female Wistar rats, even after stimulation with a current strength of 50 mA. In contrast, SVF was induced in all the wild SHR males after stimulation at thresholds of 35 mA, 25 mA, and 30 mA. A higher stimulation threshold (45 mA) was required to induce SVF in two male hairless SHRs, while one male did not develop SVF even after 50 mA stimulation. Compared to males, the female rats were less vulnerable to developing SVF regardless of strain. SVF occurred only in two wild SHR females after stimulation with the maximal current strength of 50 mA.

### 3.5. Microscopic Examination of Heart Tissue

#### 3.5.1. Assessment of Cardiac Tissue Structural Changes Using Hematoxylin–Eosin Staining

As illustrated in Figure 2A, structural changes, as observed through hematoxylin–eosin staining, were less pronounced in the left ventricle of the hairless SHR males compared to the wild SHR. Moderate alterations were observed in the right ventricles of both SHR rat strains. In contrast, the left ventricular structural changes were mild in females, regardless of whether they were wild or hairless SHR, and were not apparent in the right ventricles (Figure 2B).

#### 3.5.2. Evaluation of Collagen Deposition in Cardiac Tissue Using Van Gieson Staining

As illustrated in Figure 3A, mild collagen deposition was detected in the extracellular space of the left ventricle in both the wild and hairless SHR males compared to the sex-matched normotensive Wistar rats. Collagen staining was less pronounced in the female wild and hairless SHR rats compared to the normotensive Wistar rats (Figure 3B).

#### 3.5.3. Detection of Alkaline Phosphatase Activity

The histochemical demonstration of endothelial alkaline phosphatase (AP) activity in the arterial capillaries of the left and right ventricles in both the male and female rats is presented in Figure 4A,B. AP is involved in trans-phosphorylation which includes adenosine metabolism. It suggests a functional arterial capillary network. According to the quantitative image analysis, there were no significant alterations in AP activity in the arterial capillary network in either the left or right ventricles among the rat strains, regardless of sex (Figure 4C).

#### 3.5.4. Detection of Dipeptidyl Peptidase-4 Activity in Venous Capillaries

The histochemical demonstration of endothelial dipeptidyl peptidase-4 (DPP4) activity in the venous capillaries of the left and right ventricles in the male and female rats is presented in Figure 5A,B. According to the quantitative image analysis, there were no significant changes in DPP4 activity in the venous capillary network of either the left or right ventricles among the rat strains, regardless of sex (Figure 5C). However, there was a tendency for decreased DPP4 activity in the left ventricle of the hairless SHR males. DPP4 has been implicated in collagen metabolism and increased activity promotes fibrosis and arrhythmias [27].

#### 3.5.5. Immunofluorescent Detection of Myocardial Cx43

The immunolabeling of Cx43, used to assess its density and distribution in the cardiomyocytes of the left and right ventricles of the male and female rat strains, is illustrated in Figure 6A,B. In the normotensive rats, Cx43 is predominantly localized at the intercalated disks of cardiomyocytes, with a minimal presence on their lateral sides. In contrast, the hypertensive rat strains exhibit enhanced lateralization of Cx43, particularly in the left ventricle of males and, to a lesser extent, in the right ventricle, and in females. The quantification of total Cx43 immunolabeling and its abnormal distribution to the lateral sides of cardiomyocytes is presented in Figure 6C,D. However, according to the statistical analysis, these differences were not significant.

### 3.6. Myocardial Total tCx43 Protein Levels and Its Phosphorylated pCx43^368^ Variants

The representative immunoblots of tCx43 and pCx43^368^ with corresponding GAPDH controls for all the experimental groups are illustrated in Figure 7A. The protein expression of Cx43 in the left and right ventricles of the male and female rats from the examined strains is illustrated in Figure 7B. Compared to the normotensive rats, there was a significant decrease in the Cx43 protein levels in the left ventricles of the wild SHR males, though to a lesser extent in hairless SHR males. We detected increased Cx43 expression in the right ventricles of the male wild SHR compared to the left ventricles of the wild SHR males. There were no differences in the right ventricles among all the male strains. We detected an increase in Cx43 expression only in the right ventricle of the hairless SHR compared to the wild SHR, in females.

Parallel to this, the pCx43^368^ variant, reflecting the phosphorylation of Cx43 at serine 368, was significantly reduced in the left ventricles of the wild SHR males but not in the hairless SHR males (see Figure 7C). We observed increased Cx43 expression in the right ventricle of the wild SHR males compared to the left ventricle of the same strain. There were no differences in the right ventricles among all the male strains. In females, Cx43 expression was increased exclusively in the right ventricle of the hairless SHR compared to the wild SHR. These findings are consistent with the observed patterns in tCx43 expression.

### 3.7. Cx43 Modulating PKCε and PKCδ

The phosphorylation of Cx43 is essential for the function of Cx43 channels, ensuring electrical coupling among cardiomyocytes. Several protein kinases are involved in Cx43 phosphorylation, with PKCε and PKCδ being particularly relevant to hypertension. As shown in Figure 8B, we detected a decrease in PKCε protein expression in the wild SHR males compared to the normotensive Wistar rats. In the right ventricle, we observed increased PKCε expression in the wild SHR compared to the left ventricle of the same strain. In females, we detected increased expression in the right ventricle of the hairless SHR compared to the wild SHR. In PKCδ, we detected increased protein expression in the right ventricles of all the strains compared to the left ventricles of the same strains. There were no significant differences among the experimental groups in females (Figure 8C).

### 3.8. Extracellular Matrix Protein (TGFβ, SMAD2, and MMP-2) Alterations in Left and Right Heart Ventricles

To monitor profibrotic signaling in hypertensive rat hearts, we analyzed the relevant protein markers of the extracellular matrix. TGFβ protein levels were significantly increased in the left ventricles of the wild SHR males. In females, we detected a decrease in TGFβ protein expression in both the left and right ventricles of the hairless SHR compared to the wild SHR (Figure 9B). There were no significant changes in the SMAD2 protein levels in either the left or right ventricles among the male and female rats across strains (Figure 9C). Interestingly, MMP-2 protein levels were significantly higher in the right ventricles of both the males and females of the normotensive Wistar and wild SHR rats, but not in the right ventricle of the female hairless SHR (Figure 9D).

### 3.9. Myocardial Activity of MMP-2

Zymography was performed to detect MMP-2 activity, which is involved in modulating extracellular matrix protein levels. As shown in Figure 10B, MMP-2 activity was significantly reduced in the left ventricles of both the wild and hairless SHR males. We observed a reduction in MMP-2 activity in the right ventricle of the wild SHR males compared to the left ventricle. There were no changes in the left and right ventricles of females, regardless of strain.

Representative images of spontaneously hypertensive rats with a mutation in desmoglein 4 exhibiting a hairless phenotype are on Figure 11. 

## 4. Discussion

The findings of this study contribute to further understanding of the pathogenesis of essential or primary HTN, highlighting sex-dependent differences in myocardial Cx43 expression, as well as differences between the left and right heart ventricles. The downregulation of Cx43 and its phosphorylated variant pCx43^368^ was most pronounced in the left heart ventricles of the wild SHR males, but not in the right ventricles. Compared to males, Cx43 expression was not significantly altered in either heart ventricles in females, regardless of the rat strain. The sex-dependent differences in response to HTN in rodent models, similar to those observed in humans, may be explained by sexual dimorphisms, with females being protected by estrogen [28,29].

Adaption to a cold environment is considered cardioprotective [16,17] and likely contributes to benefits in hypertensive individuals. Indeed, the acclimation of hairless SHR males to an ambient temperature that is below thermoneutrality [19] resulted in the attenuation of Cx43 downregulation, which may contribute to their reduced susceptibility to life-threatening arrhythmias (as indicated by a higher VF threshold) and improved heart function (as indicated enhanced by an enhanced ejection fraction). Moreover, our findings highlight that HTN downregulated Cx43 in the left ventricles of males, but not in females. This may contribute to the lower vulnerability of females’ hearts to malignant arrhythmias, as demonstrated in this study.

The downregulation of Cx43 and its phosphorylated variant pCx43^368^ was even more attenuated in older, nine-month-lasting acclimation of hairless SHR males [20]. This acclimation was associated with increased thermogenesis and the metabolic adaptation of brown adipose tissue (BAT) in both males and females [19]. An increase in BAT mass in hairless SHR compared to wild SHR was also confirmed in this study. This suggests that humans suffering from HTN may benefit from cold environment adaptation due to increased BAT and adipocyte thermogenesis. The latter is promoted by sympathetic activation during cold exposure via the expression of mitochondrial uncoupling protein 1 [30].

In addition to the attenuation of Cx43 downregulation in hairless SHR males, there was a reduction in its abnormal topology, specifically an enhanced localization on the lateral sides of cardiomyocytes. The lateralization of Cx43 is highly pro-arrhythmic due to electrical instability affecting AP propagation and alterations in conduction pathways [8,11]. This results in non-uniform anisotropy, creating an arrhythmogenic substrate [9]. The remodeling process of Cx43 may be affected by pharmacotherapy [11] as well as by acclimation to environmental temperature, as demonstrated in hairless SHR males. It is important to note the significance of the early initiation of therapy [31] or cold acclimation. Additionally, hairless SHR exhibited reduced heart rate-corrected QT interval (QTc) compared to wild SHR. An increased duration of this parameter is recognized as a predictor of mortality in HTN [32,33].

Besides Cx43 alterations, myocardial remodeling, i.e., the hypertrophy of cardiomyocytes, increased collagen deposition, and fibrosis contributes to electrical instability and the development of cardiac arrhythmias and heart dysfunction in HTN [8]. In line with this, we observed widened extracellular space with polymorphonuclears in the left heart ventricles, suggesting tissue inflammation in wild SHR males, and to a lesser extent in the right heart ventricles as well as in hairless SHR. These changes were less pronounced in the left ventricles of both wild and hairless SHR females and were not observed in the right ventricles. This suggests sex- and heart chamber-related differences in response to HTN and the benefits of the acclimation of hairless SHR. The latter was also indicated by reduced pro-fibrotic signaling mediated by TGFβ protein in the left heart ventricles of hairless SHR males.

Unexpectedly, the protein levels of MMP-2, which is involved in extracellular matrix homeostasis [34], were significantly increased in the right heart ventricles of all the examined strains, regardless of sex. However, the activity of MMP-2 was significantly reduced only in the left heart ventricles of the wild SHR and hairless SHR males. This presents a challenge for further investigation since MMP-2 activity was not altered in females, regardless of heart chamber or rat strain.

Data from humans indicate that coronary microvascular dysfunctions in hypertrophied hearts contribute to heart failure [35,36]. In this context, we examined AP activity involved in adenosine metabolism and purinergic signaling in arterial capillaries, as well as DPP-4 activity in venous capillaries, which is known to be involved in collagen metabolism and arrhythmias [27]. According to the QIA results, there were no significant changes in AP activity among the rat strains, regardless of sex or heart ventricles, suggesting the preserved function of arterial capillaries. However, there was a tendency for DPP-4 activity to decline in the left ventricles of hairless SHR males, which may impact collagen metabolism in extracellular space [37].

Altogether, our data suggest that reduced Cx43 expression, along with its mislocalization in structurally remodeled hearts, may contribute to the increased propensity of males to malignant arrhythmias and the progression of heart dysfunction in HTN. On the other hand, the suppression of these deleterious pro-arrhythmic factors through increased thermogenesis in acclimated hairless SHR males reduces arrhythmogenicity and susceptibility to ventricular fibrillation, while also improving ejection fraction and end-systolic volume. Noteworthy, females and the right heart ventricles of males were much less responsive to HTN.

In summary: Objective: Spontaneously hypertensive rat (wild SHR) is a relevant model of essential hypertension in humans resulting from the interactions of multiple genetic and environmental factors predisposing to cardiac arrhythmias and heart failure. The hairless SHR strain is a model characterized by increased thermogenesis and metabolic adaptations due to chronic acclimation to an ambient temperature that is below thermoneutral.

Questions: Are there differences in the expression of myocardial electrical coupling protein Cx43 between wild SHR versus hairless SHR? Are there differences in Cx43 expression between the left and right heart ventricles of hypertensive rats? Is there sex-related difference in cardiac Cx43 expression? Is there an impact of Cx43 expression on the vulnerability of the heart to malignant ventricular arrhythmias?

Key findings: We demonstrated that the expression of myocardial Cx43 was significantly reduced solely in the left ventricle of wild SHR males, while to a lesser extent, in hairless SHR. It may contribute to a lower vulnerability of the heart to malignant arrhythmias as estimated by increased VF threshold versus wild SHR. Noteworthy, Cx43 expression was not significantly altered in the right heart ventricles of either rat strain and in both heart ventricles of females that were much less susceptible to inducible VF.

Meaning: The findings indicate that increased thermogenesis due to adaptation to the environmental condition may attenuate the pro-arrhythmic downregulation of myocardial Cx43 in essential hypertension and protect from life-threatening cardiac arrhythmias.

It would be possible to expect the benefits of chronic adaptation to a mild cold environment for individuals suffering from primary hypertension to hamper the propensity of the heart to malignant arrhythmias and progression to mechanical dysfunction.

Concluding remarks: Hypertension is a multifactorial disease influenced by genetic, epigenetic, and environmental components [38]. It would be valuable to further investigate the roles of miR-1 and miR-206, which modulate cardiac Cx43 expression and contribute to arrhythmogenesis [39,40], potentially in the context of HTN as well. Additionally, the impact of environmental temperature on the human epigenome is an important area for understanding temperature-related health effects [41].

The limitations of this study include the inability to establish a causal relationship between the cold acclimation-induced upregulation of Cx43 and propensity to cardiac arrhythmias. Additionally, the smaller number of rats examined for VF threshold, due to the limited availability of hairless SHR, could also be considered a limitation.

## Figures and Tables

**Figure 1 biomolecules-14-01509-f001:**
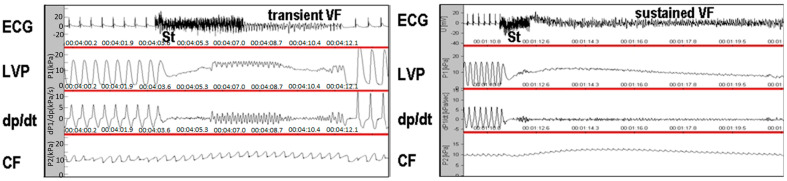
Illustration of recording parameters in an ex vivo perfused rat heart used for electrical burst stimulation (St) to estimate the incidence of sustained ventricular fibrillation (SVF). Parameters include LVP (left ventricular developed pressure), dp/dt (rate of pressure development, an index of contractility and relaxation), and CF (coronary flow).

**Figure 2 biomolecules-14-01509-f002:**
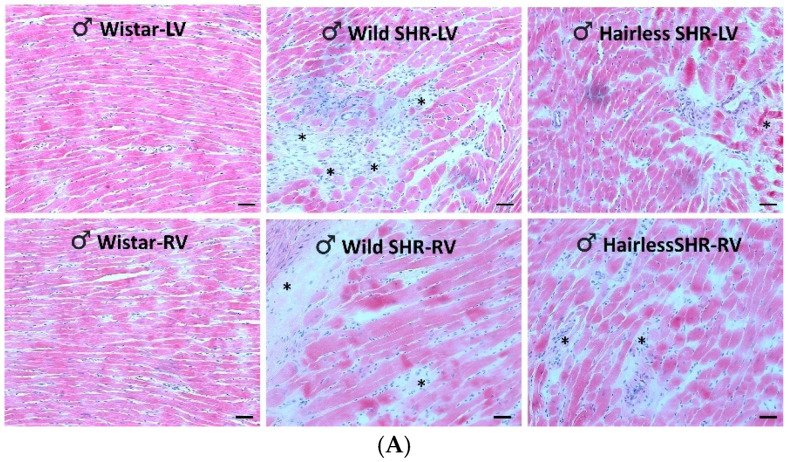
(**A**). Representative HE images of male rats show apparent myocardial structural alterations in the left ventricle (LV) and, to a lesser extent, in the right ventricle (RV) of both the wild SHR and hairless SHR, *n* = 6. Asterisks denote polymorphonuclear cells in the widened extracellular space. Scale bar = 100 µm. (**B**). The representative HE images of female rats show moderate myocardial structural alterations in the left ventricle (LV) and minor alterations in the right ventricle (RV) of both wild SHR and in the left ventricle of hairless SHR. *n* = 6. Asterisks denote polymorphonuclear cells in the widened extracellular space. Scale bar = 100 µm.

**Figure 3 biomolecules-14-01509-f003:**
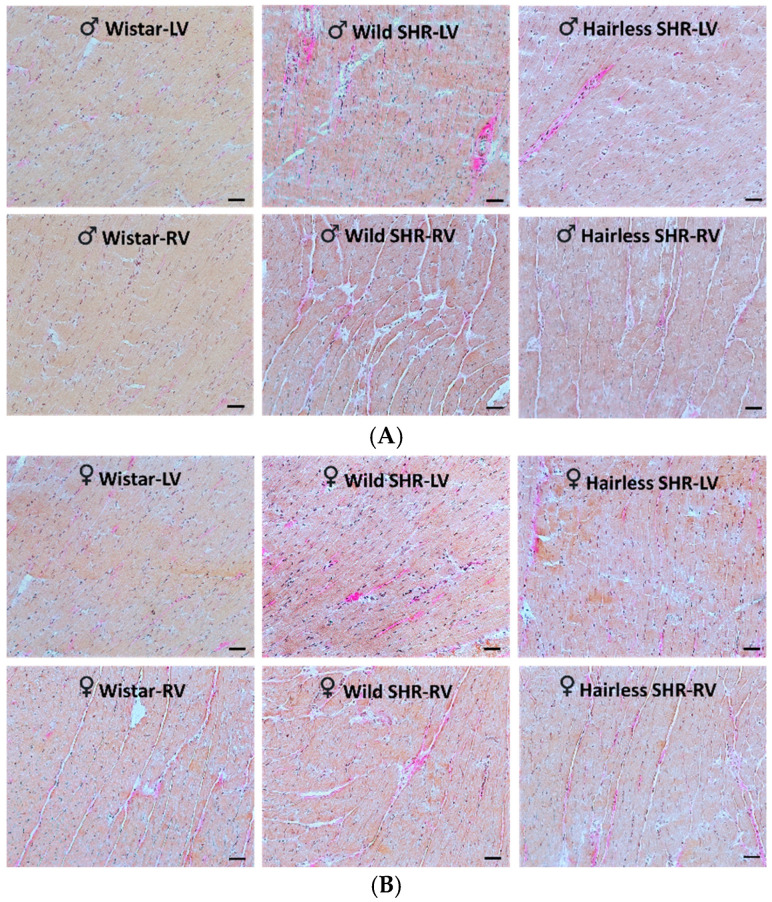
(**A**). Representative VG images of male rats show mild collagen deposition (pink) in the left (LVs) and, to a lesser extent, in the right ventricles (RVs) of both the wild and hairless SHRs compared to the normotensive Wistar rats. *n* = 6. Scale bar = 100 µm. (**B**). The representative VG images of the female rats show minor changes in extracellular collagen deposition (pink) in the left (LVs) and, to a lesser extent, in the right ventricles (RVs) of both the wild and hairless SHRs compared to the normotensive Wistar rats. *n* = 6. Scale bar = 100 µm.

**Figure 4 biomolecules-14-01509-f004:**
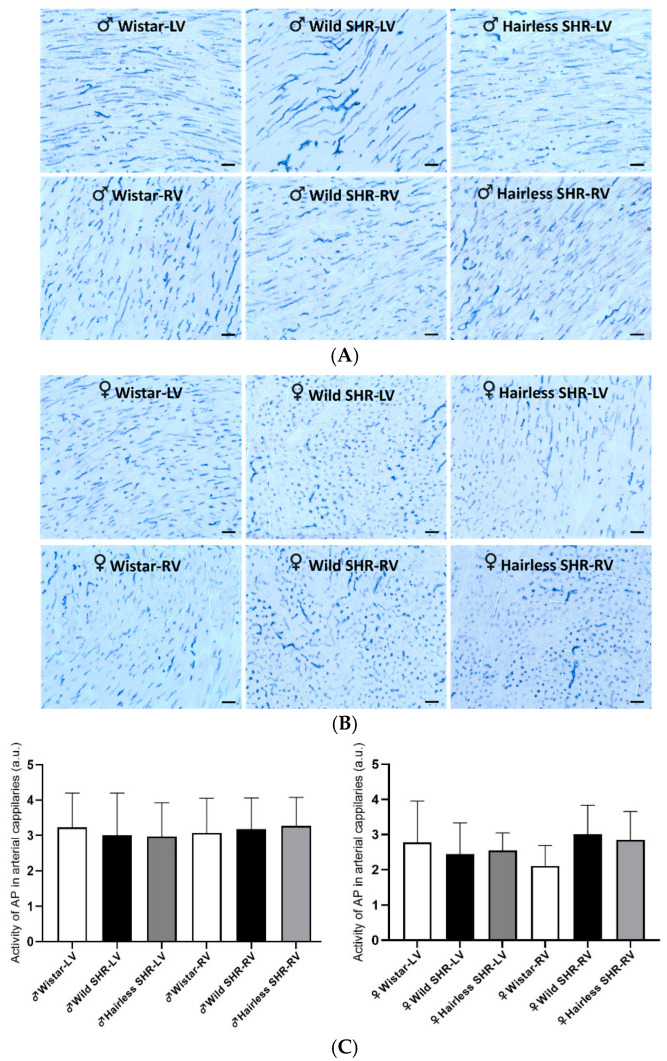
(**A**). Representative light microscopy images showing alkaline phosphatase activity (blue) in the endothelial cells of arterial capillaries in the left (LVs) and right ventricles (RVs) of the male hairless SHR, wild SHR, and normotensive Wistar rats are presented. Note the high AP activity in the heart tissue across all the examined rat strains, regardless of sex. *n* = 6 per group. Scale bar = 100 µm. (**B**). Representative light microscopy images display alkaline phosphatase activity (blue) in the endothelial cells of arterial capillaries within the left (LVs) and right ventricles (RVs) of the male rats, including the hairless SHR, wild SHR, and normotensive Wistar strains. The images highlight the high AP activity observed in the heart tissue of all the rat strains, irrespective of sex. *n* = 6 per group. Scale bar = 100 µm. (**C**). The quantitative image analysis of endothelial alkaline phosphatase (AP) activity, a functional marker of arterial capillaries, was performed on the experimental rats. No significant differences were observed among the rat strains in either the left (LVs) or right ventricles (RVs), regardless of sex. *n* = 6 per group. Data are presented as means ± SD.

**Figure 5 biomolecules-14-01509-f005:**
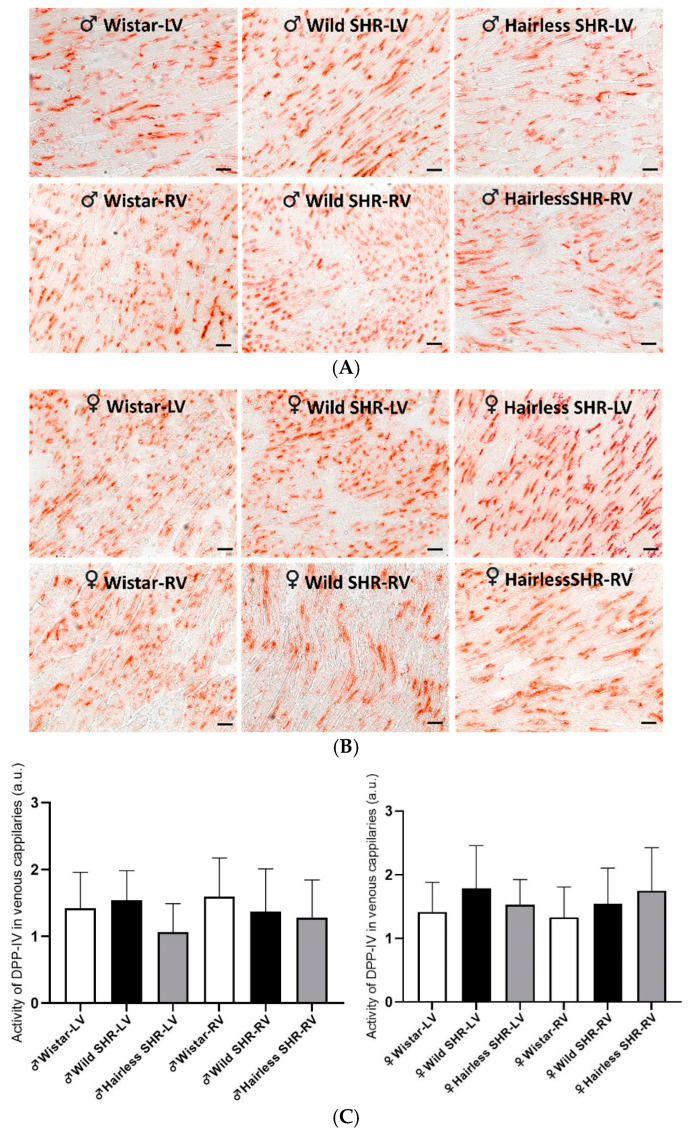
(**A**). Representative light microscopy images show dipeptidyl peptidase-4 (DPP4) activity (brown color) in the venous capillaries of the experimental rats. Note the high activity in the left (LVs) and right ventricles (RVs) of all the male rat strains, with less pronounced activity in the left ventricle of the hairless SHR males. *n* = 6 per group. Scale bar = 100 µm. (**B**). Representative light microscopy images display dipeptidyl peptidase-4 (DPP4) activity (brown color) in the venous capillaries of the experimental rats. High DPP4 activity is observed in the left (LVs) and right ventricles of both the wild and hairless SHRs. *n* = 6 per group. Scale bar = 100 µm. (**C**). The quantitative image analysis of dipeptidyl peptidase-4 (DPP4) activity, a functional marker of venous capillaries, was conducted in the left (LVs) and right ventricles (RVs) of the experimental rats. Overall, there were no significant differences among the rat strains in either the left or right ventricles, regardless of sex. However, a tendency toward decreased DPP4 activity was observed in the left ventricle of the hairless SHR males. *n* = 6 per group. Data are presented as means ± SD.

**Figure 6 biomolecules-14-01509-f006:**
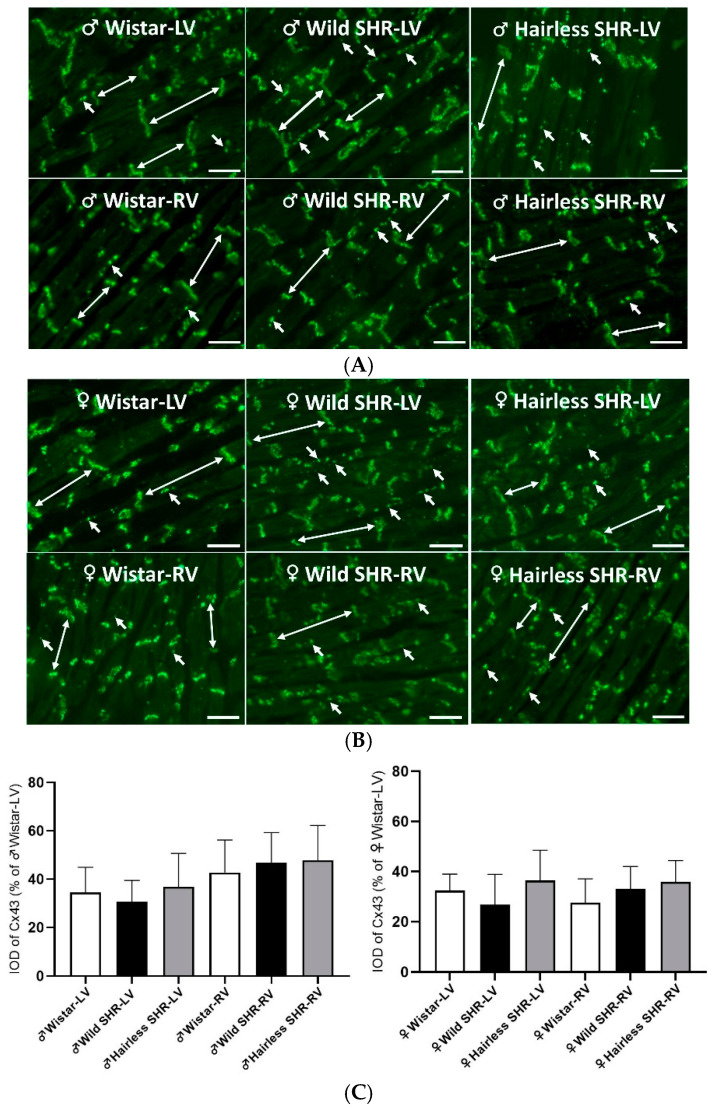
(**A**). Immunofluorescence labeling of Cx43 was used to detect its density and distribution in the left (LVs) and right ventricles (RVs) of the normotensive and hypertensive male rats. Note the predominant localization at the intercalated disks (long arrows), which is typical in normotensive hearts, and the enhanced lateral localization of Cx43 (short arrows) in the hypertensive strains. *n* = 6 per group. Scale bar = 100 µm. (**B**). The immunofluorescence labeling of Cx43 was used to assess its density and distribution in the left (LVs) and right ventricles (RVs) of the normotensive and hypertensive female rats. Note the predominant localization at the intercalated disks (long arrows), which is typical in normotensive hearts, and the enhanced lateral localization of Cx43 (short arrows) observed in the hypertensive strains. In females, there is less pronounced lateral localization across all the hypertensive strains compared to males. *n* = 6 per group. Scale bar = 100 µm. (**C**). The quantitative image evaluation of the integral optical density (IOD) of total Cx43 did not reveal significant differences between the left (LVs) and right ventricles (RVs) in the male or female hypertensive rats compared to the normotensive rats. However, there was a tendency toward a decline in Cx43 expression in the left, but not the right ventricles of the wild SHR males, as well as in both the ventricles of hairless SHR. *n* = 6 per group. Data are presented as means ± SD. (**D**). The quantitative image analysis of Cx43 topology on the lateral sides of cardiomyocytes showed a clear tendency for increased lateral distribution of Cx43 in the left ventricles (LVs) of the wild SHR males compared to the normotensive strain, with a lesser extent observed in the hairless SHR males. The lateralization was less pronounced in the right ventricles (RVs) of males and in both ventricles of females. However, differences among the groups were not significant. *n* = 6 per group. Data are presented as means ± SD.

**Figure 7 biomolecules-14-01509-f007:**
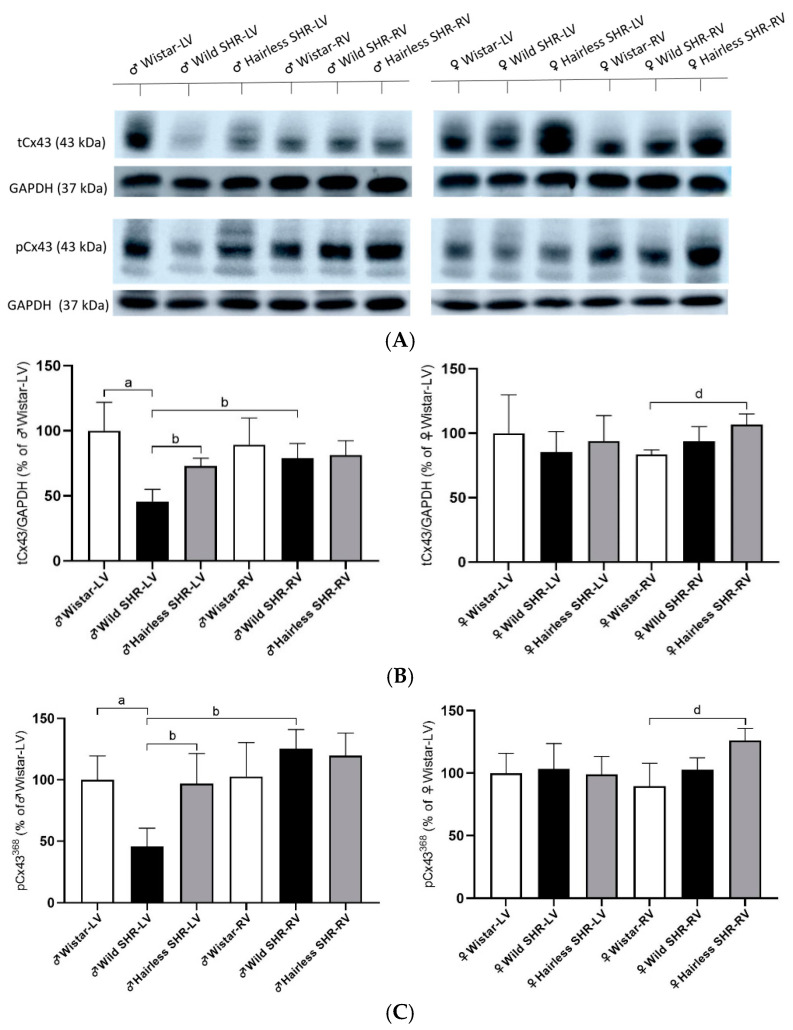
(**A**). Representative immunoblots of tCx43 and pCx43^368^ with corresponding GAPDH controls for all the experimental groups. (**B**). Cx43 protein levels in the left (LVs) and right ventricles (RVs) of the males and females from the examined rat strains were determined by Western blotting. Significant reduction in Cx43 protein was noted in the left ventricles of the wild SHR, with a lesser reduction in the hairless SHR. In contrast, there were no differences in the right ventricles among all the male strains. In females, Cx43 expression was increased only in the right ventricle of the hairless SHR compared to the wild SHR. *n* = 6 per group. Data are presented as means ± SD. ^a^ *p* < 0.05 vs. ♂ Wistar-LV, ^b^ *p* < 0.05 vs. ♂ SHR-LV, and ^d^ *p* < 0.05 vs. ♀ Wistar-RV. For statistical evaluation, one-way ANOVA and Bonferroni´s post hoc tests were used. (**C**). The protein levels of the pCx43^368^ variant were notably reduced in the left ventricles (LVs) of the wild SHR males but not in the hairless SHR. pCx43^368^ expression was higher in the right ventricle (RV) of the wild SHR males compared to the left. In females, Cx43 expression was increased only in the right ventricle of the hairless SHR versus the wild SHR; *n* = 6 per group. Data are presented as means ± SD. ^a^ *p* < 0.05 vs. ♂ Wistar-LV, ^b^ *p* < 0.05 vs. ♂ SHR-LV, and ^d^ *p* < 0.05 vs. ♀ Wistar-RV. For statistical evaluation, one-way ANOVA and Bonferroni´s post hoc tests were used. The original images of Western blot can be found in Appendix A.

**Figure 8 biomolecules-14-01509-f008:**
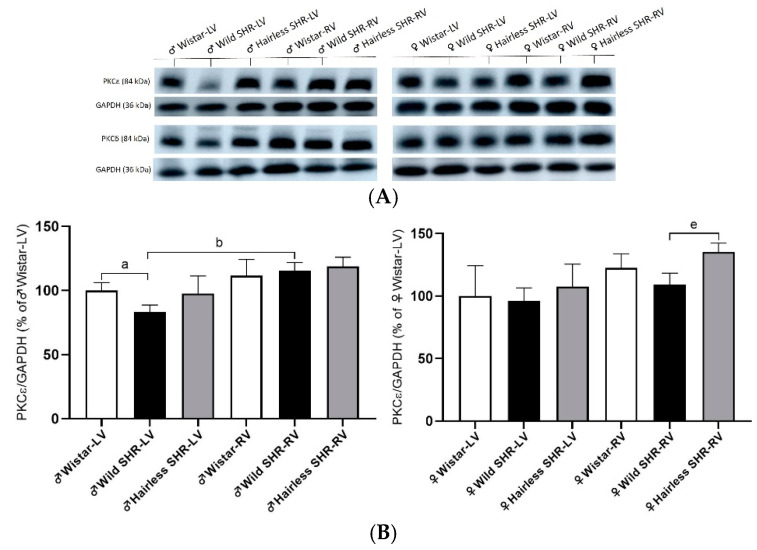
(**A**). Representative immunoblots of PKCε and PKCδ with corresponding GAPDH controls for all the experimental groups. (**B**). The protein expression of PKCε was significantly reduced in the left ventricle (LV) of the male wild SHR compared to the normotensive Wistar strain. In the right ventricle (RV), PKCε expression was higher in the wild SHR compared to the left ventricle of the same strain. In females, the right ventricle of the hairless SHR had higher expression than the wild SHR. *n* = 6 per group. Data are presented as means ± SD. ^a^ *p* < 0.05 vs. ♂ Wistar-LV, ^b^ *p* < 0.05 vs. ♂ SHR-LV, and ^e^ *p* < 0.05 vs. ♀ SHR-RV. For statistical evaluation, one-way ANOVA and Bonferroni´s post hoc tests were used. (**C**). The protein expression of PKCδ was higher in the right ventricles (RVs) of all the strains compared to the left (LVs). *n* = 6 per group. Data are presented as means ± SD. ^a^ *p* < 0.05 vs. ♂ Wistar-LV and ^b^ *p* < 0.05 vs. ♂ SHR-LV. For statistical evaluation, one-way ANOVA and Bonferroni´s post hoc tests were used. The original images of Western blot can be found in Appendix A.

**Figure 9 biomolecules-14-01509-f009:**
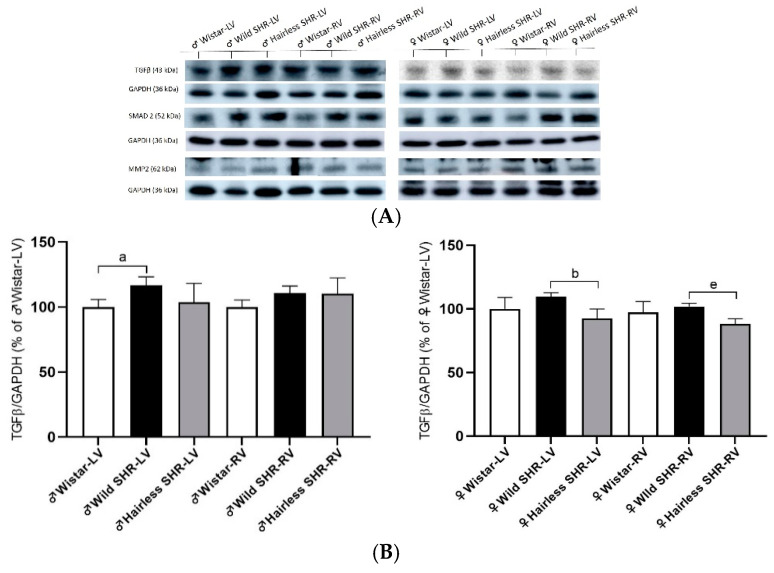
(**A**). Representative immunoblots of TGFβ, SMAD2, and MMP-2 with corresponding GAPDH controls for all the experimental groups. (**B**). TGFβ protein expression was significantly increased in the left heart ventricles (LVs) of the wild SHR males. In females, TGFβ protein expression was decreased in both ventricles of the hairless SHR compared to the wild SHR. *n* = 6 per group. Data are presented as means ± SD. ^a^ *p* < 0.05 vs. ♂ Wistar-LV, ^b^ *p* < 0.05 vs. ♂ SHR-LV, and ^e^ *p* < 0.05 vs. ♀ SHR-RV. For statistical evaluation, one-way ANOVA and Bonferroni´s post hoc tests were used. (**C**). There were no significant changes in the SMAD2 protein levels either in the left (LVs) or right heart ventricles (RVs) among rat strains of males as well as females. *n* = 6 per group. Data are presented as means ± SD. For statistical evaluation, one-way ANOVA and Bonferroni´s post hoc tests were used. (**D**). MMP-2 levels were higher in the right ventricles (RVs) of both the males and females of the normotensive Wistar and wild SHR rats, but not in the female hairless SHR. Data are presented as means ± SD. ^a^ *p* < 0.05 vs. ♂ Wistar-LV, ^b^ *p* < 0.05 vs. ♂ SHR-LV, and ^c^ *p* < 0.05 vs. ♂ Hairless SHR-LV. For statistical evaluation, one-way ANOVA and Bonferroni´s post hoc tests were used. The original images of Western blot can be found in Appendix A.

**Figure 10 biomolecules-14-01509-f010:**
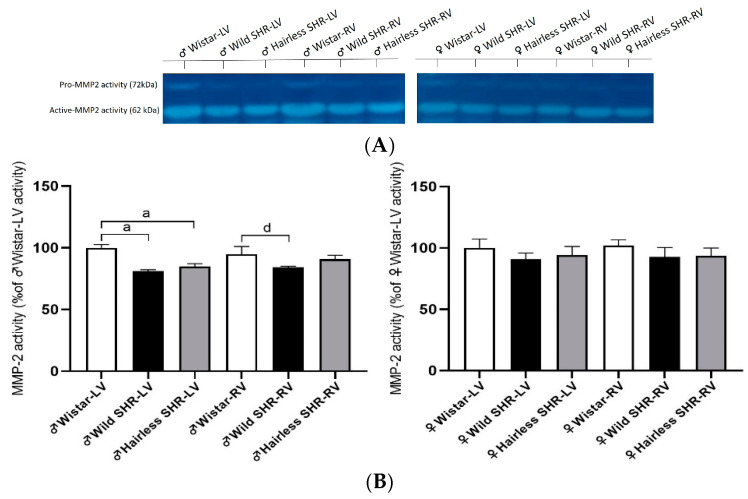
(**A**). Representative zymograms of MMP-2 activity. (**B**). Note the significant decline in MMP-2 activity in the left ventricles (LVs) of the wild and hairless SHR males. MMP-2 activity was reduced in the right ventricle (RV) of the wild SHR males compared to the left. Data are presented as means ± SD. ^a^ *p* < 0.05 vs. ♂ Wistar-LV and ^d^ *p* < 0.05 vs. ♀ Wistar-RV. For statistical evaluation, one-way ANOVA and Bonferroni’s post hoc tests were used. The original images of Western blot can be found in Appendix A.

**Figure 11 biomolecules-14-01509-f011:**
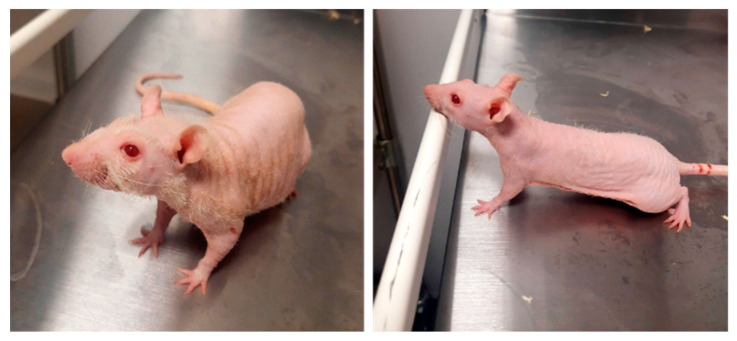
Spontaneously hypertensive rats with a mutation in desmoglein 4 exhibiting a hairless phenotype, left—male, right—female (unpublished photographs).

**Table 1 biomolecules-14-01509-t001:** Biometric parameters of experimental rats.

	BP (mm Hg)	BW (g)	HW (g)	LVW (g)	RVW (g)	BAT (g)	BAT/BW
♂ Wistar	128.6 ± 11.2	388.5 ± 40.2	1.41 ± 0.06	0.79 ± 0.09	0.23 ± 0.05	0.31 ± 0.08	0.76 ± 0.11
♂ Wild SHR	162.9 ± 13.8 ^a^	340 ± 21.4	1.47 ± 0.20	0.91 ± 0.10	0.25 ± 0.08	0.33 ± 0.04	0.97 ± 0.11
♂ Hairless SHR	165.0 ± 16.9 ^a^	300.1 ± 23.1 ^a^	1.36 ± 0.05	1.00 ± 0.07	0.25 ± 0.03	0.42 ± 0.06 ^b^	1.42 ± 0.29 ^ab^
♀ Wistar	120.3 ± 27.7	286.4 ± 55.6 ^a^	0.88 ± 0.08 ^a^	0.58 ± 0.09	0.18 ± 0.05	0.24 ± 0.05	0.89 ± 0.27
♀ Wild SHR	152.4 ± 11.5 ^d^	197.5 ± 37.7 ^bd^	0.86 ± 0.05 ^b^	0.68 ± 0.10 ^b^	0.16 ± 0.03 ^b^	0.21 ± 0.06 ^b^	1.04 ± 0.22
♀ Hairless SHR	150.2 ± 10.0 ^d^	180.1 ± 10.3 ^cd^	0.98 ± 0.06 ^c^	0.78 ± 0.12 ^cd^	0.19 ± 0.04	0.27 ± 0.09 ^c^	1.46 ± 0.03 ^d^

BP—blood pressure (mm Hg); BW—body weight (g); HW—heart weight (g); LVW—left ventricular weight (g); RVW—right ventricular weight (g); BAT—brown adipose tissue (g). *n* = 6. Data are presented as means ± SD; ^a^ *p* < 0.05 vs. ♂ Wistar, ^b^ *p* < 0.05 vs. ♂ Wild SHR, ^c^ *p* < 0.05 vs. ♂ Hairless SHR, and ^d^ *p* < 0.05 vs. ♀ Wistar. For statistical evaluation, one-way ANOVA and Bonferroni´s post hoc tests were used.

**Table 2 biomolecules-14-01509-t002:** Echocardiographic heart structural parameters.

	IVSd (mm)	LVPWd (mm)	LVIDd (mm)	LVIDs (mm)	RWT (mm)
♂ Wistar	1.37 ± 0.16	1.49 ± 0.11	6.81 ± 0.40	4.28 ± 0.40	0.44 ± 0.04
♂ Wild SHR	1.77 ± 0.10 ^a^	1.67 ± 0.09	7.77 ± 0.25 ^a^	5.17 ± 0.10 ^a^	0.43 ± 0.03
♂ Hairless SHR	1.73 ± 0.15	1.81 ± 0.25	7.32 ± 0.20	4.21 ± 0.37 ^b^	0.49 ± 0.07
♀ Wistar	1.61 ± 0.08	1.48 ± 0.12	7.12 ± 0.05	4.06 ± 0.21	0.41 ± 0.01
♀ Wild SHR	1.86 ± 0.08	1.55 ± 0.22	6.44 ± 0.42 ^b^	3.81 ± 0.39 ^b^	0.49 ± 0.10
♀ Hairless SHR	1.84 ± 0.30	1.66 ± 0.04	6.52 ± 0.21	3.59 ± 0.22	0.51 ± 0.01

IVSd (mm)—interventricular septal thickness in diastole; LVPWd (mm)—left ventricular posterior wall thickness in diastole; LVIDd (mm)—left ventricular internal diameter in diastole; LVIDs (mm)—left ventricular internal diameter in systole; RWT—relative wall thickness (RWT = [(LVPWd + IVSd)/LVIDd]). *n* = 6. Data are presented as means ± SD; ^a^ *p* < 0.05 vs. ♂ Wistar and ^b^ *p* < 0.05 vs. ♂ Wild SHR. For statistical evaluation, one-way ANOVA and Bonferroni′s post hoc tests were used.

**Table 3 biomolecules-14-01509-t003:** Echocardiographic heart functional parameters.

	HR (beat/min)	CO (L/min)	EF (%)	FS (%)	EDV (mL)	ESV (mL)
♂ Wistar	459 ± 6	0.25 ± 0.06	76.25 ± 3.18	37.60 ± 4.39	0.72 ± 0.11	0.17 ± 0.05
♂ Wild SHR	334 ± 45 ^a^	0.23 ± 0.02	67.83 ± 2.08 ^a^	33.50 ± 1.80	1.04 ± 0.09 ^a^	0.33 ± 0.02 ^a^
♂ Hairless SHR	353 ± 41 ^a^	0.25 ± 0.04	79.00 ± 4.77 ^b^	42.83 ± 4.19 ^b^	0.90 ± 0.05	0.19 ± 0.05 ^b^
♀ Wistar	377 ± 12 ^a^	0.24 ± 0.01	79.50 ± 2.12	43.25 ± 2.47	0.81 ± 0.12	0.17 ± 0.03
♀ Wild SHR	330 ± 21	0.17 ± 0.03	77.00 ± 2.65 ^b^	40.83 ± 4.36	0.62 ± 0.11 ^b^	0.14 ± 0.04 ^b^
♀ Hairless SHR	383 ± 23	0.22 ± 0.04	82.50 ± 2.65	46.50 ± 2.64	0.64 ± 0.06 ^c^	0.11 ± 0.01

HR (beat/min)—heart rate; CO (l/min)—cardiac output; EF (%)—ejection fraction; FS (%)—fraction shortening; EDV (ml)—end-diastolic volume; ESV (ml)—end-systolic volume. *n* = 6. Data are presented as means ± SD; ^a^ *p* < 0.05 vs. ♂ Wistar, ^b^ *p* < 0.05 vs. ♂ Wild SHR, ^c^ *p* < 0.05 vs. ♂ Hairless SHR. For statistical evaluation, one-way ANOVA and Bonferroni´s post hoc tests were used.

**Table 4 biomolecules-14-01509-t004:** Electrocardiographic parameters of experimental rats.

	PQ (ms)	QRS (ms)	QT (ms)	QTc
♂ Wistar	28.65 ± 6.35	20.54 ± 4.35	56.69 ± 7.59	72.56 ± 9.02
♂ Wild SHR	30.90 ± 7.64	20.22 ± 6.25	67.50 ± 8.36	90.35 ± 8.64 ^a^
♂ Hairless SHR	27.00 ± 3.30	17.67 ± 1.89	57.36 ± 6.71	62.83 ± 10.32 ^b^
♀ Wistar	28.86 ± 5.82	17.48 ± 1.81	54.58 ± 2.92	77.40 ± 17.39
♀ Wild SHR	32.48 ± 5.58	16.28 ± 2.18	65.80 ± 6.19	75.26 ± 21.88
♀ Hairless SHR	35.50 ± 12.20	21.50 ± 2.12	63.25 ± 2.47	82.01 ± 21.21

PQ (ms)—interval between the start of the P wave and the start of the Q wave; QRS (ms)—the time interval of the duration of the QRS complex; QT (ms)—interval measures the time from the beginning of the Q wave to the end of the T wave; QTc—corrected time interval between the Q wave and the T wave, adjusted for heart rate. *n* = 6. Data are presented as means ± SD; ^a^ *p* < 0.05 vs. ♂ Wistar and ^b^ *p* < 0.05 vs. ♂ Wild SHR. For statistical evaluation, one-way ANOVA and Bonferroni´s post hoc tests were used.

**Table 5 biomolecules-14-01509-t005:** Electrical threshold to induce SVF in experimental rats.

	Number of Rats	Current Magnitude (mA)	SVF
♂ Wistar	3	50	No
♂ Wild SHR	1	35	Yes
	1	25	Yes
	1	30	Yes
♂ Hairless SHR	2	45	Yes
	1	50	No
♀ Wistar	3	50	No
♀ Wild SHR	2	50	Yes
	1	50	No
♀ Hairless SHR	3	50	No

Sustained ventricular fibrillation (SVF) was induced at 35, 25, and 30 mA current strength in wild SHR males, while at higher 45 mA in two hairless SHRs and none in one rat. Females were less prone to develop SVF even at max 50 mA stimulation regardless of rat strain. *n* = 3 per group.

## Data Availability

Data are contained within the article and Appendix A.

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
