# Peer review of "Acclimation of Hairless Spontaneously Hypertensive Rat to Ambient Temperature Attenuates Hypertension-Induced Pro-Arrhythmic Downregulation of Cx43 in the Left Heart Ventricle of Males"

_biomolecules, 2024, doi:10.3390/biom14121509_

Round 1
Reviewer 1 Report
Comments and Suggestions for Authors
In the manuscript by Andelova K et al. entitled “Acclimation of hairless SHR to ambient temperature attenuates hypertension induced pro-arrhythmic down-regulation of Cx43 in the left heart ventricle of males”, the authors describe the effects of acclimation of hairless SHR rats compared to wild SHR and normosensitive strains analysing different factors that increase susceptibility to malignant arrhythmias. Specifically, authors investigate Cx43, pCx43 and ECM proteins evidencing the differences in their expression between the left and the right heart ventricles both in males and females rats.
The manuscript by Andelova et al. addresses an important topic such as hypertension associated to malignant arrhythmias and cardiovscular diseases which are an urgent scientific challenge to be met.
The work sounds scientifically good. It is well performed from a methodological point of view and the results are described and discussed coherently also regarding the current literature. However, the results are only partially innovative compared to what has already been described in previous works. Moreover, as reported by the authors themselves in the concluding remarks of the paper, it was not possible to establish a causal relationship between the Cx43 increase due to cold acclimatization and cardiac arrhythmias. Figures are of good quality but several issues to be addressed are listed below.
1. In fig 2A the second picture in the upper raw is representative of Wild-SHR-LV, isn’t it?
2. In fig 3A and 3B the collagen expression is shown. Quantitative image analyses of collagen expression could be usefull to evaluate even slight differences among strains or between males and females.
3. In fig 6B short arrows dimension should be uniformed.
4. In the 3.6 paragraph (lane 446) the authors stated that “we detected an increase in Cx43 expression only in the right ventricle of hairless SHR compared to wild SHR, in females” but in the blot image (fig 7A) the increase of Cx43 is evident also in hairless SHR-LV compared to wild SHR-LV, in females. Can you explain please? If this expression is not significative, as reported on the graph of fig 7B, the blot image should be changed to avoid misunderstandings.
5. In all figures representing blots (7A and B; 8A and B; 9A) and in the one representing the zymograms of MMP-2 (fig 10A) typing errors occurred because only Wistar-RV is reported for both males and females samples.
6. In all figure captions p value should be more clearly specified for an immediate comprehension.
7. Lane 484: please specify that the decrease of PKCε protein expression was observed in the left ventricle.
8. In fig 9B does bp value refer to males or females?
9. In fig 10A caption the p values described are not correspondent to those indicated on the graph bars.
10. Lane 563: please verify the sentence “regarless the sex”. Is it correct?
11. In both manuscript and original figures blot molecular weights shoul be reported.
12. In the original MMP-2 blot of males, no Wistar-RV heart samples is run: can you explain?
13. In the original figures only two Hairless SHR-LV or -RV samples are blotted compared to the three Wild SHR-LV or -RV samples both in males and in females, but in the figure captions the authors state the use of n=6 per group. Can you clarify? Moreover, the two samples show a very different band intensity regarding some of the antigens investigated and the criterion for choosing to represent one band rather than the other seems a little arbitrary. Please explain.
Reviewer 2 Report
Comments and Suggestions for Authors
Aim of this manuscript is to deep dive into beneficial effects of mild hypotermia on connexin and ECM proteins in both hairless and non-hairless rats as well as investigate other cardiac parameters.
My concerns are as follows:
- all rats were housed at 22°C hence it is not evident the beneficial effect of lower temperature compared to RT. The very same experimental set shall be performed on rats housed at standard conditions to allow an effective comparison of the results obtained.
- All western blots original images lack of GADPH so it is not possible to evaluate the proteins expression in a correct way. Please run all samples providing images for both protein and GAPDH for all proteins analysed; this will allow to report correct protein expression result normalized for GAPDH. This shall be done for each protein under investigation.
- Datra in fig. 4C, 5C, 6C, 6D lack of statistical analysis.
Reviewer 3 Report
Comments and Suggestions for Authors
General comments
The study shows that reduced electrical coupling protein Cx43 expression, along with its mislocalization in structurally remodeled hearts, may contribute to the increased propensity of males to malignant arrhythmias and the progression of heart dysfunction in hypertension.
On the other hand, the suppression of these deleterious pro-arrhythmic factors through Females and the right heart ventricles of males were much less responsive to hypertension.
The study appears to have appropriate methodology. The data are clearly presented. The manuscript is generally well written, but some issues need to be clarified.
1. In the title as well as in the abstract and discussion authors are writing about acclimation. What does it mean? In the Materials and Methods section there is no information about acclimation. According to the authors the rats were housed in 22°C (line 90), which is an ideal temperature for these animals, so from this point of view the animals do not undergo acclimation.
2. In the Materials and Methods section there is no information about the age of the animals, as well as about the duration of the experiment. It must be completed.
3. Why from the groups of 9 animals (line 84) in the Results section there are groups of 6 animals? What did happen with the remaining 3 animals? A suspicious reader might think that maybe these lost animals did disturb the statistics….
4. In the case of MMP-2 activity the authors should provide information which band was analyzed: proMMP2 (about 72 kDa) or active MMP2 (about 68 kDa).
Reviewer 4 Report
Comments and Suggestions for Authors
The study by Katarina Andelova, Matus Sykora, Veronika Farskasova, Tatiana Stankovicova, Barbara Szeiffova Bacova, Vladimir Knezl, Tamara Egan Benova, Michal Pravenec and Narcis Tribulova name “Acclimation of hairless SHR to ambient temperature attenuates hypertension induced pro-arrhythmic down-regulation of Cx43 in the left heart ventricle of males” is natural continuation cycle works team authors researching molecular mechanisms development hypertension and the role of temperature adaptation in their implementation. In this work, the authors compared the role of the connexin 43-dependent signaling cascade in the development of hypertension depending on sex, strain, temperature conditions and, most importantly, on specific structures of the heart (the left and right ventricles were compared heart). There are numerous works devoted to the contribution of connexin 43 to the development of cardiovascular pathologies, which indicates the undeniable relevance of the field. However, the authors managed to find their own direction of research with a possible practical outcome. It increases the value of the present work. The authors have carried out a large amount of experiments. The results are presented clearly and consistently. The conclusions are logical and based on the results. However, there are several issues that require clarification before publication.
Line 90: Unfortunately, it is not clear from the available description of the experimental conditions how exactly the SHR adaptation was performed. It is necessary to add the experimental scheme and describe in more detail the temperature conditions for different groups of animals and the time the animals were kept in the specified conditions. Photos of the animals' habits are also desirable.
line 125: If I understand correctly, the ECG recording system used by the authors allows changing the ECG in several leads simultaneously. Since the quantitative results of the ECG analysis depend on the standard lead used, the authors would be well advised to indicate which of the standard leads they analyzed.
line 164: It would be desirable to indicate the incubation time with antibodies and how non-specific binding of antibodies was blocked.
line 170: The values used for the illuminant brightness settings, exposure time, binning should be specified. What background brightness gamma correction or other image enhancement/post-processing methods were used?
Classical quantitative brightness analysis is usually attempted in gray scale to minimize artifacts of RGB filters SSD matrix of a digital camera. Authors are encouraged to provide links to relevant reference and give a more detailed description of the method for calculating the RGB threshold, and justify the chosen threshold value.
At the authors' discretion, I would advise them to review the algorithm for processing color photographs. For example, selecting from the RGB image only the color channel of interest (blue for AP, the sum/average of green and red for DPP4), as well as automated selection of regions of interest (ROIs) using the threshold discrimination method. I believe that the choice of the target channel and adequate selection of ROIs should reduce the variation of values, allowing a more adequate assessment of the differences between the experimental groups. Operations for working with individual colors in an RGB image are easily implemented in most programs, including the freely distributed ImageJ.
Line 217. Classically, the tests used to assess the normality of distribution are first described then the criteria used to test hypotheses.
Table 3: It is desirable to add figures with representative fragments of ECG recordings, at least for the groups between which statistical differences were found.
The authors should also analyze QT dispersion as a classically analyzed ECG characteristic at hypertension (doi: 10.3389/fmed.2020.583331)
In the discussion, the authors would be encouraged to include a summary diagram of their results: the mechanism of hypertension development and the effect of thermal adaptation on associated signaling events.
Can the authors make lifestyle recommendations for patients with arrhythmias/hypertension in the discussion?
line 84 It should be (n=8)
Authors are encouraged to decipher abbreviations when first used. The abbreviation connexin 43 must be deciphered.
Best regards
Round 2
Reviewer 1 Report
Comments and Suggestions for Authors
I have no comments or suggestions.
Reviewer 2 Report
Comments and Suggestions for Authors
Thank you, all concerns are cleared from my side
Reviewer 4 Report
Comments and Suggestions for Authors
The manuscript was significantly revised. All comments were taken into account. Answers to all questions of interest are provided.
Best regards